# Corrosion Performance Analysis of Tubing Materials with Different Cr Contents in the CO$_2$ Flooding Injection–Production Environment

Xuehui Zhao [1,2,*], Guoping Li [3], Junlin Liu [3], Mingxing Li [4], Quanqing Du [5] and Yan Han [1,2]

1    State Key Laboratory for Performance and Structure Safety of Petroleum Tubular Goods and Equipment Materials, CNPC Tubular Goods Research Institute, Xi'an 710077, China
2    Key Laboratory of Petroleum Tubular Goods and Equipment Ouality Safety for State Market Regulation, Xi'an 710077, China
3    Research Institute of drilling and Production Technology of Qinghai Oilfield, Dunhuang 736202, China
4    Oil and Gas Engineering Research Institute of Petro China Changqing Oilfield Company, Xi'an 710077, China
5    The No.4 Production Plant, Qinghai Oilfield, Mangya 816400, China
*    Correspondence: zhaoxuehui@cnpc.com.cn

**Abstract:** In order to clarify the difference in corrosion performance between low Cr-containing (3Cr, 5Cr, and 9Cr) tubing material and carbon steel N80 in the Carbon dioxide (CO$_2$) flooding injection and production environment and the range of adaptation, corrosion tests and analysis were carried out in simulated working conditions. In this paper, the electrochemical potentiodynamic testing technology and the weight loss method were used to comparatively analyze the corrosion performance and variation law of three types of tubing materials with different Cr contents in a simulated CO$_2$ flooding-produced water environment under different partial pressure conditions. Additionally, scanning electron microscopy and Energy Dispersive Spectrometer (EDS) analysis were conducted to examine the surface corrosion morphology characteristics and elemental composition of material films under various conditions. The results indicate that the open circuit potentials of 3Cr, 5Cr, and carbon steel N80 were similar under the same experimental conditions. However, the open circuit potentials of 9Cr were relatively high and there was an obvious passivation zone in anodic polarization. Nevertheless, compared to that of 13Cr, the passivation state was unstable, and pitting corrosion continued to expand once it formed. This demonstrates that the corrosion resistance of the material can be effectively enhanced and a stable passivation state can be achieved in the anodic polarization region when the Cr content of the material reaches at least 13%. The service life of materials can be predicted based on their corrosion rate under high temperature and pressure simulation environments. We found that 9Cr materials exhibited good adaptability while 3Cr and 5Cr materials showed poor adaptability. Therefore, it was not recommended to use 3Cr and 5Cr materials. Therefore, 3Cr, 5Cr, and N80 materials will be used at lower partial pressure levels of CO$_2$ (<0.2 MPa).

**Keywords:** carbon dioxide flooding; electrochemistry; corrosion rate; low Cr-containing tubing

## 1. Introduction

Corrosion has always been the focus of oil and gas field exploration, development, and deep exploitation. Every year, serious accidents, environmental pollution, and casualties caused by the corrosion failure of oil casing pipes result in significant economic losses and have a negative impact on the country [1,2]. Especially in recent years, with the proposed national "dual carbon" emission reduction target, carbon dioxide capture, utilization, and storage (CCUS), as a new technology with significant potential for large-scale emission reduction, has been increasingly applied and popularized to varying degrees in major oilfields [3–5]. However, many problems such as pipe corrosion and fracture failure, which

affect the efficiency of $CO_2$ flooding, pose major obstacles to promoting the widespread adoption of this technology. Therefore, proposing efficient corrosion prevention and control technology and successfully applying it in oilfields is one of the key means to slow down pipe corrosion [6,7]. Currently, various anticorrosion measures are used in oilfields, including anticorrosion materials, coatings, corrosion inhibitors, etc. [8–11]. Considering the economic cost, it is expected to explore an anticorrosion method that can meet the production cycle and have a cost-effective investment in $CO_2$ environments for oilfields [12–16]. Therefore, it is recommended to use relatively economical pipes with certain corrosion resistance in oilfields, such as different low-alloy pipes with varying Cr contents (3Cr, 5Cr, 9Cr, etc.) [17–20]. However, due to the complexity of the oil and gas field environment, especially the coupling effect of corrosion factors such as the harshness of the fluid in a $CO_2$ flooding injection–production environment and variations in $CO_2$ concentration in associated gas, there are many controversies and uncertainties regarding the adaptability of these materials in an oilfield environment. Meng et al. [21] studied the corrosion behavior of 3Cr steel under the condition of oil–water two-phase laminar flow, and the results showed that the corrosion rate of the material reached 3.56 mm/a in the single-water-phase environment of the oilfield and the corrosion rate was 1.6 mm/a in the water phase containing oil. Gu Lin et al. [22] studied the corrosion behavior of 3Cr pipes in production wells containing oxygen flooding, and the results showed that when $O_2$ (3%) and $CO_2$ (4.01%) coexisted in production wells, $O_2$ played a significant catalytic role in $CO_2$ corrosion. With an increase in oxygen content, the corrosion rate of pipes increased sharply, and the corrosion rate of 3Cr was much higher than that of an extremely severe corrosion grade. Zhao Guoxian et al. [23] studied the high temperature and high pressure corrosion characteristics of 5Cr casing steel under different $CO_2$ partial pressures and showed that the depth and diameter of pitting pits on the surface did not change significantly with the change in $CO_2$ partial pressure from low to high under a high temperature and high pressure corrosion environment while the pitting rate showed a gradually decreasing trend. Zhang Siqi et al. [24] studied the corrosion performance of 3Cr steel under a coexisting $CO_2/H_2S$ environment and showed that the corrosion resistance of 3Cr steel under a coexisting environment is relatively better than that under a $CO_2$ environment. Ji Nan et al. [25] analyzed the corrosion causes of 3Cr steel tubing in a gas injection well and showed that the inner and outer walls of 3Cr steel tubing were mainly corroded by dissolved oxygen, resulting in a mismatch between the properties of tubing materials and the actual service corrosion environment. Xia Wenbin et al. [26] analyzed the reasons for the cracking of a L80-3Cr corrosion-resistant oil well pipe end and showed that a brittle bainite structure was produced in the process of pipe making, resulting in poor cracking resistance in the material. The research results of these studies have no reference significance in a $CO_2$ flooding injection–production environment. This paper mainly analyzes the adaptability of three kinds of low-Cr oil pipe materials in a $CO_2$ flooding injection–production environment in order to provide a reliable data basis for oilfield material selection. So, based on the injection–production environment of $CO_2$ flooding in XX Oilfield, the corrosion performance and adaptability of three types of tubing materials with different Cr contents are evaluated and analyzed in this paper. The study explored the differences in and adaptability range of corrosion performance for Cr-containing materials in the produced water environment and various corrosion systems in XX Oilfield. It also clarified whether these three types of Cr-containing materials meet the practical application requirements, providing a positive scientific basis and technical support for the optimal selection of downhole tubing strings in the injection–production environment of $CO_2$ flooding in XX Oilfield.

## 2. Experimental Procedures

### 2.1. Test Material

The test material is a commercial tubing material used in the oilfield, and its chemical composition (mass fraction) is shown in Table 1. The metallographic structure of the tubing material, mainly consisting of martensitic structure, is depicted in Figure 1.

**Table 1.** Chemical composition of material for test (wt%).

| Element | C | Si | S | P | Mn | Cr | Ni | Mo | V | Cu | Ti |
|---------|------|------|-------|-------|------|------|-------|-------|-------|--------|--------|
| 3Cr | 0.25 | 0.25 | 0.002 | 0.009 | 0.48 | 2.91 | 0.065 | 0.076 | 0.012 | 0.024 | 0.003 |
| 5Cr | 0.091 | 0.20 | 0.002 | 0.009 | 0.36 | 4.90 | 0.045 | 0.09 | 0.021 | 0.053 | 0.002 |
| 9Cr | 0.12 | 0.31 | 0.005 | 0.014 | 0.36 | 9.01 | 0.066 | 1.00 | 0.017 | 0.0098 | 0.0001 |
| N80 | 0.25 | 0.36 | 0.005 | 0.010 | 1.41 | 0.21 | 0.016 | 0.004 | / | 0.015 | / |

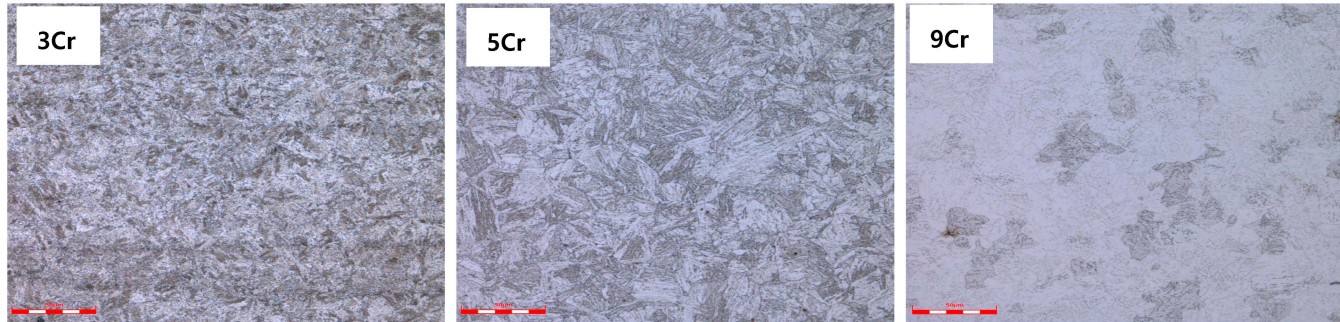

**Figure 1.** Metallographic structure analysis of three kinds of Cr-containing materials.

### 2.2. Experimental Methods

#### 2.2.1. Electrochemical Test

Potentiodynamic polarization curves were measured using the M273 potentiostat produced by American PerkinElmer Company and its 352 SoftCorr III software test system. A three-electrode system was adopted, with the sample serving as the working electrode. The silver chloride electrode (Ag-AgCl) was used as the reference electrode while a graphite rod was used as an auxiliary electrode. The scanning rate of the moving electrode was set at 0.3 mV/s. During the test, the working electrode was a square sample with an area of 1 cm$^2$. The other side was welded with copper wire, and all non-working faces were coated with epoxy resin. The working face of the sample was polished using 600#–1000# SiC water-based sandpaper, washed with distilled water after oil removal using acetone, and dried for later use. The simulation test parameters were carried out according to the most harsh environment of the XX Oilfield, and the test solution medium was prepared in the laboratory according to the ion concentration provided by the oilfield (refer to Table 2 for details of each ion concentration). The test temperature was set at 80 °C, which simulates harsh conditions in wellbores. Additionally, $CO_2$ gas with a concentration $\geq$ 99.99% was used as the corrosive agent during testing.

**Table 2.** Ion concentration of solution medium in laboratory simulation test (mg/L).

| Name | BaCl$_2$ | Na$_2$SO$_4$ | NaHCO$_3$ | CaCl$_2$ | NaCl |
|------|----------|--------------|-----------|----------|---------|
| Concentration | 1976.4 | 167.56 | 157.92 | 848.0 | 23882.0 |

#### 2.2.2. High Temperature and High Pressure Immersion Test

The High Temperature and High Pressure (HTHP) autoclave was used to simulate the formation water environment produced in the oilfield, and we compared and analyzed the corrosion performance and pitting corrosion sensitivity of three low Cr tubing materials. The sample size was 40 mm × 10 mm × 3 mm, and the test solution was prepared according to Table 2. Prior to testing, nitrogen was introduced into the solution for at least two hours to deoxidize it. After installing the sample, nitrogen deoxidization continued for an additional thirty minutes. Then, $CO_2$ gas with varying partial pressures was introduced based on the test conditions and heated to the required temperature before timing began. The test

period lasted for 168 h. Following testing, any residual corrosive medium on the surface of the sample was washed away using distilled water. Then, after drying off any remaining water from its surface, corrosion product film removal took place followed by weighing of weight loss via FR2300MK electronic balance in order to calculate average corrosion rate of said sample. Finally, VE-GA II scanning electron microscope (SEM) observation occurred regarding surface corrosion morphology while analytically pure chemical reagents and gases were used throughout. The schematic diagram of the high-temperature and high-pressure soaking test device is shown in Figure 2.

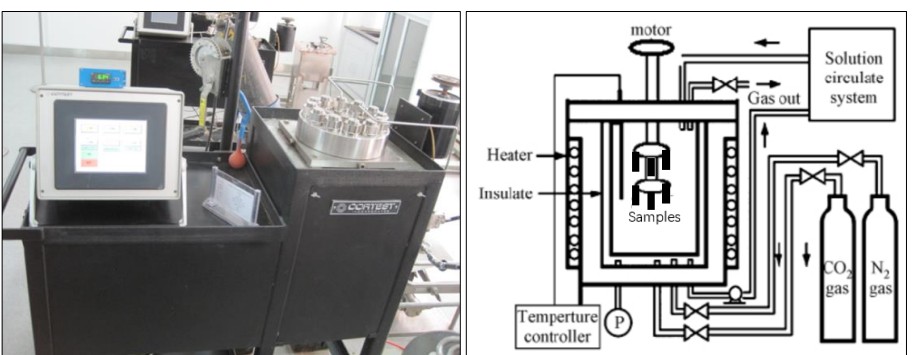

**Figure 2.** Diagram of high-temperature and high-pressure soaking test device.

## 3. Results and Discussion

### 3.1. Electrochemical Corrosion Behavior of Materials

Figure 3 illustrates the open circuit potential curves of 3Cr, 5Cr, and 9Cr materials in saturated $CO_2$ solution medium. Comparing the open circuit potentials of the three materials ($E_{3Cr}$ = −671 mV, $E_{5Cr}$ = −675 mV, and $E_{9Cr}$ = −627.5 Mv), it can be seen that under the same test conditions, the open circuit potentials of 3Cr and 5Cr materials are basically similar. However, the open circuit potential of the 9Cr material is relatively higher by approximately ΔE = 43 mV compared to the first two materials. This indicates that the corrosion activity of the 9Cr material is comparatively weaker in this particular solution medium. The increase in Cr content is helpful for the formation of a $Cr(OH)_3$ passivation film on the surface [27,28], and 9Cr material has relatively good stability and corrosion resistance compared to the other two materials containing low Cr. The open circuit potential when using carbon steel N80 as the contrast material is $E_{N80}$ = −683 mV, which is close to the potential of the first two Cr-containing materials.

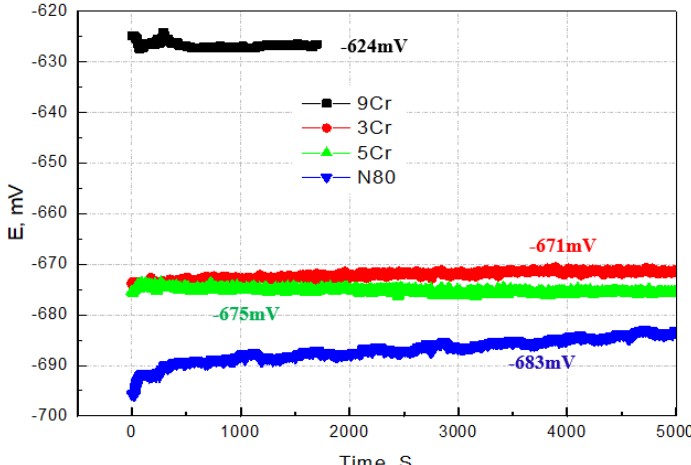

**Figure 3.** Open circuit potential curves of tubing materials under the same simulation environment.

Figure 4 illustrates the polarization curves of 3Cr, 5Cr, and 9 Cr materials at different $CO_2$ concentrations and 80 °C, including a 50% $CO_2$ aqueous solution (Condition 1) and a saturated $CO_2$ aqueous solution (Condition 2). It can be observed from the polarization curve in Figure 4 that the self-corrosion potential of the material containing 9% chromium is relatively high under both concentrations of $CO_2$, indicating a weak thermodynamic tendency for corrosion in the test environment [29]. At the same time, the anode curve has obvious passivation zone, and the passivation current density remains unchanged or decreases, which shows that it is easy for the surface of the material to be deactivated and the film hinders the further corrosion of the material. However, when the anodic polarization potential increases continuously and reaches the pitting corrosion explosion potential, the passivation film breaks down, resulting in a sudden increase in corrosion current density ($I_{corr}$) and subsequent pitting corrosion. Comparing the polarization curves and curve indicators of three Cr-containing materials under Condition 2 (Table 3), it can be seen that the self-corrosion potential of the materials under Condition 2 has a slight upward trend, which indicates that the surface passivation film is easy to form under this environment, so that the thermodynamic kinetic energy of the materials increases, thus weakening the corrosion trend. At the same time, the curve slightly shifts to the right and the corrosion current density increases relatively. From the dynamic analysis, once corrosion occurs, the corrosion rate is relatively large, and the self-corrosion potentials of 3Cr and 5Cr materials in the two $CO_2$ environments are similar to those of N80, which indicates that the corrosion resistance of these two materials containing low Cr is similar to that of carbon steel N80. Compared with the chemical compositions of the materials, the Cr content only increases by 2%–4%, not enough to obviously improve the corrosion resistance of the materials in this simulated test environment.

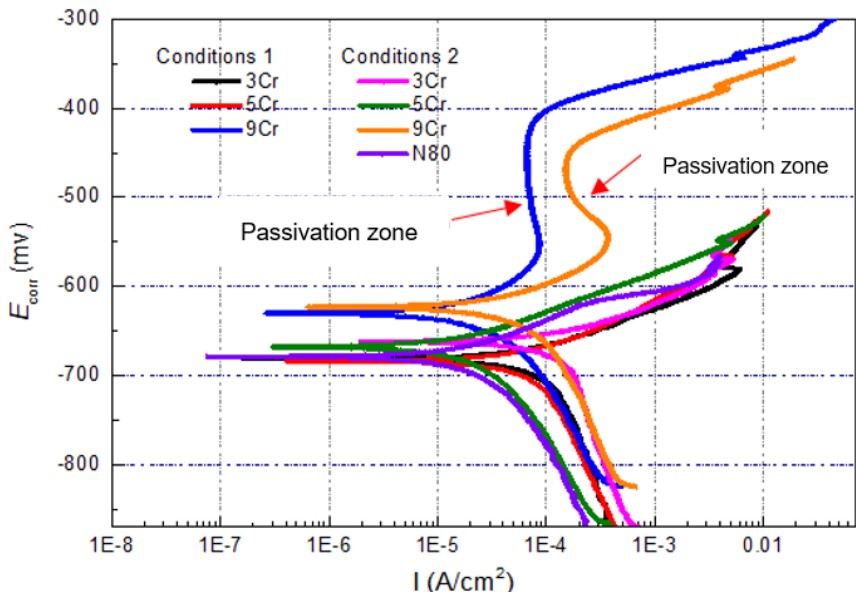

**Figure 4.** Test results of polarization curves of materials containing low Cr under different conditions.

**Table 3.** Electrochemical performance indicators of materials in different conditions.

| Material | N80 | 3Cr | 5Cr | 9Cr | N80 | 3Cr | 5Cr | 9Cr |
|---|---|---|---|---|---|---|---|---|
| | | $E_{corr}$ | | | | $I_{corr}$ | | |
| Condition 1 | / | −680 | −683 | −630 | / | $2.47 \times 10^{-6}$ | $2.68 \times 10^{-6}$ | $1.56 \times 10^{-6}$ |
| Condition 2 | −678 | −661 | −667 | −625 | $5.32 \times 10^{-6}$ | $1.04 \times 10^{-5}$ | $1.59 \times 10^{-6}$ | $4.05 \times 10^{-6}$ |

Figure 5 shows the cyclic anodic polarization curve of Cr-containing materials in a simulated oilfield environment in which 13Cr, with a relatively high Cr content, is used as the contrast material. The cyclic polarization curve can judge the local corrosion tendency

of materials by encapsulating the hysteresis envelope area. It can be seen from the test results that there is an intersection point $\varphi_{rp}$ (protection potential) between the forward and reverse polarization curves of 13Cr, and there is a stable passivation zone below $\varphi_{rp}$, where the pitting corrosion formed will stop developing and turn into a passivation state. It can be seen from the cyclic anodic polarization curve of the 9Cr material that the intersection point of the forward and reverse polarization curves is not in the anodic passivation zone (red circle), which indicates that although the surface passivation occurs, the passivation state is unstable, and the initial pitting corrosion continues to expand, indicating that the pitting corrosion sensitivity is high [30,31]. Under the same conditions, 3Cr and 5Cr have no passivation behavior. From a comparison of the cyclic anodic polarization curves of four Cr-containing materials, it can be seen that when the content of Cr is at least 13%, it is easy for a stable passivation zone to form on the surfaces of the materials and the pitting corrosion has a self-repairing effect to slow down the further expansion of local corrosion.

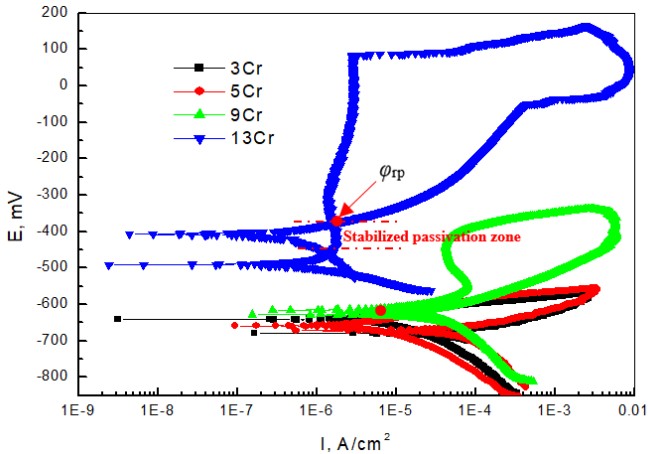

**Figure 5.** Test results of cyclic anodic polarization curves of different Cr-containing materials.

### 3.2. Effect of CO₂ Partial Pressure on Corrosion Performance of Materials

The high temperature autoclave was used to simulate $CO_2$ flooding-produced water in an oilfield and different $CO_2$ partial pressure environments, and the corrosion performance and adaptability of three kinds of Cr-containing materials under service conditions were compared and analyzed. The simulation test conditions are shown in Table 4. After each test, the sample was taken out, the residual corrosive medium on the surface was washed away with distilled water, and it was dehydrated and dried with absolute alcohol. After the test, the macroscopic and microscopic corrosion morphology were observed, and then, the average corrosion rate of the samples was calculated by using the weight loss method.

**Table 4.** Condition parameters of HTHP simulation test.

| Test Conditions | CO₂ Partial Pressure (MPa) | Temperature (°C) | Material | Medium | Test Period (h) |
|---|---|---|---|---|---|
| Condition 1 | 0.2 | | 3Cr | | |
| Condition 2 | 0.5 | 80 | 5Cr | Simulated formation water (See Table 2) | 168 |
| Condition 3 | 0.8 | | 9Cr | | |
| Condition 4 | 1 | | N80 | | |

Figure 6 shows the average corrosion rates of the materials under different test conditions, obtained by using the weight loss method after the high temperature and high pressure simulation test. It can be seen that the average corrosion rates of different Cr-containing materials are obviously different. With the gradual increase in $CO_2$ partial

pressure, the average corrosion rates of the three materials all increase by different degrees. Among them, the corrosion rates of 9Cr materials are relatively the smallest under different $CO_2$ partial pressures, and when $P_{CO2}$ is less than 1MPa, the average corrosion rates of 9Cr materials do not increase obviously and the corrosion that occurs is mild corrosion. While $P_{CO2}$ = 1 MPa, the corrosion is moderate corrosion; however, the average corrosion rates of 3Cr and 5Cr materials increase obviously, and both reach the extremely serious corrosion degree [32]. At the same time, it can be seen that the corrosion degree of carbon steel N80 is almost the same as that of 3Cr and 5Cr under the same test conditions, which shows that the corrosion resistance of 3Cr and 5Cr materials is not obviously different from that of carbon steel N80 under the simulated test conditions. Thus, the corrosion resistance of materials cannot be effectively improved when the contents of alloying element Cr are 3% and 5%.

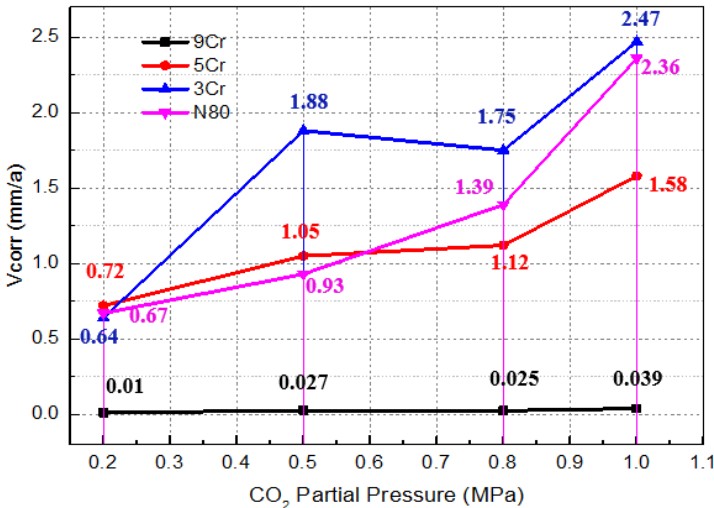

**Figure 6.** Variation trend of average corrosion rate of materials under different test conditions.

Figure 7 shows the micro-corrosion morphology of different Cr-containing materials under the environment of the high temperature and high pressure simulation test Condition 1 in which the surfaces of 3Cr and 5Cr materials are obviously corroded and the corrosion product films are rough with local shedding and discontinuous, indicating that the adhesion of the product films is poor [33,34]. The surface of the 9Cr material is flat without obvious corrosion products, which indicates that the corrosion resistance of the 9Cr material is relatively good under the condition of relatively low $CO_2$ partial pressure.

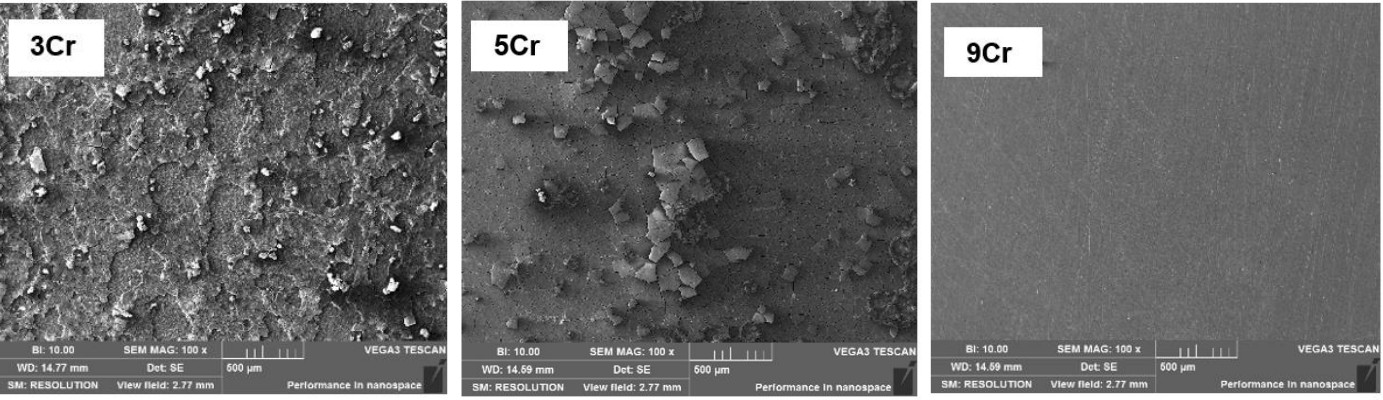

**Figure 7.** Microscopic corrosion morphology of different materials under the condition $P_{CO2}$ = 0.2 MPa.

Figure 8 shows the micro-corrosion morphology of different Cr-containing materials under the high temperature and high pressure Condition 2. The comparison shows that the corrosion product films on the surfaces of 3Cr and 5Cr materials are cracked, indicating that each product film has a certain thickness and that the internal stress in the process of water loss causes the film to crack so that it is easy for the solution medium to penetrate into the substrate through the gap and cause further corrosion in the deep layer. The surface of the 9Cr material is flat without an obvious local corrosion phenomenon under the same observation multiple, which shows that the surface film is relatively thin and the internal stress is too small during the dehydration process to cause film cracking.

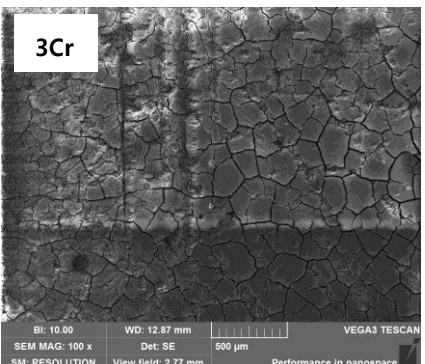 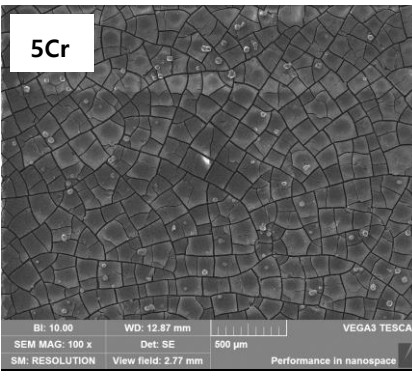 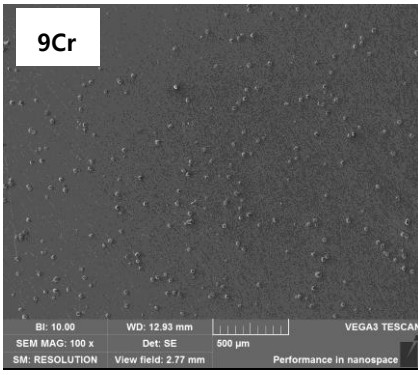

**Figure 8.** Microscopic corrosion morphology of different materials under the condition $P_{CO_2}$ = 0.5 MPa.

When increasing the $CO_2$ partial pressure, the microscopic corrosion morphologies of the materials under the simulated Conditions 3 and 4 are basically similar. As shown in Figure 9, the cracking degree of the film on the surfaces of 3Cr and 5Cr materials is increased compared with that on the surfaces of 3Cr and 5Cr materials, which shows that the increase in $CO_2$ partial pressure promotes the corrosion aggravation of the materials and the thickening of the film. In addition, two layers of cracked corrosion film can also be seen in the local shedding area (as shown in Figure 10), indicating that as long as there is a crack in the film and the matrix is exposed, the corrosive solution medium will penetrate into the surface of the substrate and rapidly react with the matrix material, resulting in the further corrosion of the material. Therefore, the density of the corrosion product film and the good adhesion with the surface of the material have a great influence on the corrosion resistance of the material. In comparison, the surface film of the 9Cr material is uniform and smooth, and there is no obvious local corrosion.

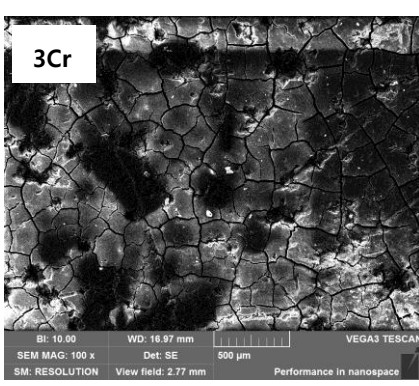 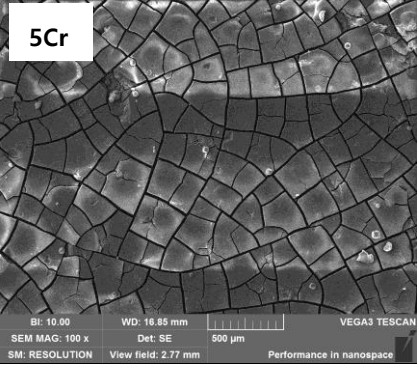 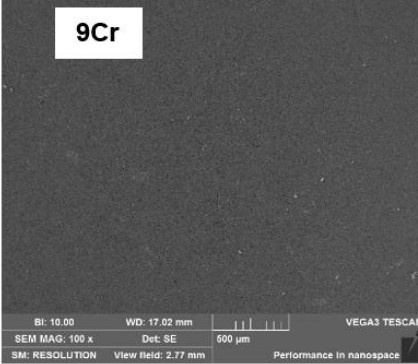

**Figure 9.** Microscopic corrosion morphology of different materials under the condition $P_{CO_2}$ = 0.8–1 MPa.

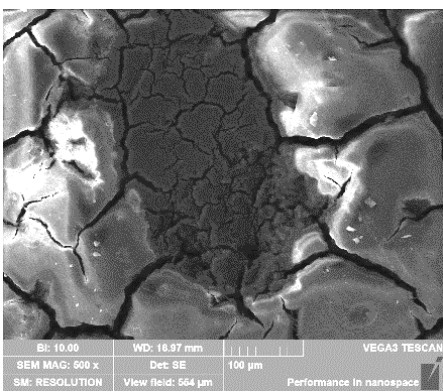

**Figure 10.** Microscopic corrosion morphology of local shedding area under the condition $P_{CO2} = 1$ MPa.

The EDS energy spectrum of the corrosion product film under different test conditions was quantitatively measured by using an energy spectrometer and the local film shedding area and Cr element enrichment on the surfaces were comparatively analyzed. The schematic diagram of EDS energy spectrum detection points is shown in Figure 11, and the test results of the element compositions of the corrosion product film under different conditions are shown in Table 5. It can be seen that the element compositions of the corrosion product film are basically similar; it is mainly composed of the Cr, O, and Fe elements. The corrosion products can be inferred as $Cr(OH)_3$, $FeCO_3$, and $Cr_2O_3$ [35]. Comparing the Cr content in the product films in Table 5, it can be seen that the Cr content in the corrosion films of 3Cr and 5Cr materials is at least three times that in the substrate. With an increase in the $CO_2$ partial pressure, the Cr content in the product films is as high as six times that of the substrate, indicating an aggravation of the corrosion degree and a strengthened Cr enrichment in the corrosion product films. However, the Cr content in the product film of the 9Cr material is about 1.5 times that of the substrate, which indicates that the product film is relatively thin. Therefore, combined with the comprehensive analysis of the corrosion rate, corrosion morphology, and Cr enrichment content in the product film, although Cr is the alloy element, it is comprehensively analyzed that the Cr enrichment on the surfaces of 3Cr and 5Cr materials is higher and the corrosion film is thicker. However, the cracking and shedding of the film do not hinder or alleviate the further corrosion of the materials. The relative enrichment ratio of the 9Cr product film is relatively low, and the corrosion product film is relatively thin, but the film is flat and compact, which alleviates the further corrosion of the material. Thus, the material has relatively good corrosion resistance. It can be seen from the energy spectrum of the shedding area of the product film that it is mainly composed of the Cr, O, and Fe elements. Hence, the bare surface of the sample is corroded again to form a secondary product film.

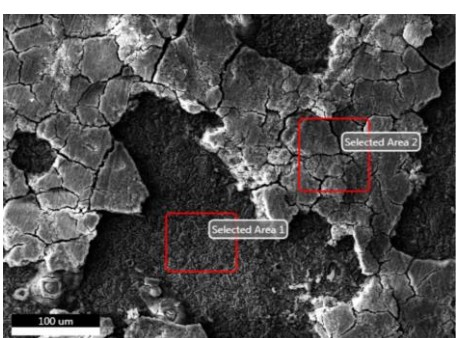

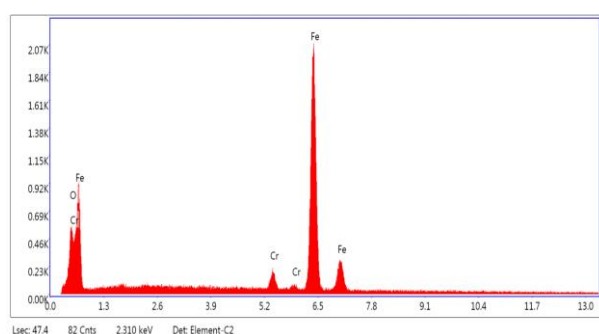

**Figure 11.** Schematic diagram of sampling points for EDS energy spectrum analysis of sample surface.

**Table 5.** EDS results of different corrosion areas on sample surfaces.

| CO₂ Partial Pressure | Element | 3Cr | | 5Cr | | 9Cr |
|---|---|---|---|---|---|---|
| | | Product Film | Shedding Area | Product Film | Shedding Area | Product Film |
| 0.2 MPa | O K | 53.25 | 18.44 | 56.97 | 27.85 | 32.74 |
| | Cr K | 11.9 | 3.77 | 19.73 | 6.99 | 10.68 |
| | Fe K | 31.83 | 77.79 | 19.83 | 65.16 | 54.67 |
| 0.5 MPa | O K | 67.23 | 52.68 | 58.12 | / | 45.67 |
| | Cr K | 20.78 | 23.66 | 32.09 | / | 15.16 |
| | Fe K | 5.52 | 17.52 | 3.88 | / | 36.38 |
| 1 MPa | O K | 69.52 | 61.38 | 71.31 | 37.96 | 46.69 |
| | Cr K | 19.17 | 23.69 | 21.38 | 6.91 | 15.22 |
| | Fe K | 5.21 | 7.28 | 3.17 | 55.13 | 35.92 |

*3.3. Effect of Corrosion Rate on Service Life of Materials*

To a great extent, the operating state of an in-service tubing string is evaluated via residual strength. So, the limit state is reached when the residual strength of the tubing string decays to a certain extent as a result of corrosion defects or damage. Thus, its residual life also reaches the limit value. In this paper, the internal yield pressure that the material can bear after corrosion thinning damage is calculated, or whether the corrosion allowance needed to meet the internal pressure meets the requirements of the design life is judged, based on the average corrosion rate of the above indoor simulation test and referring to the calculation method of the internal yield pressure of the material pipe in standard of SY/T 6328-1997. The calculation results are shown in Table 6. The service design conditions of the field tubing string are as follows: (a) The design service life of the injection–production well tubing string is 10 years. (b) Tubing specification: Φ 88.9 × 6.45 mm. (c) The maximum safe internal pressure of the tubing string during operation shall be taken as the reference according to 60 MPa (pressure balance is not considered). The pressure calculation formula is shown in Formula (1):

$$P = 0.875 \times ((2Yp \times t)/D) \tag{1}$$

*P*—minimum internal yield pressure, *Y*p—specified minimum yield strength of material, *t*—engineering wall thickness, *D*—nominal outer diameter.

**Table 6.** Service life, calculated from corrosion rate.

| Material | 9Cr | 5Cr | 3Cr | N80 |
|---|---|---|---|---|
| Average corrosion rate (mm/a) | 0.01–0.039 | 0.72–1.58 | 0.64–2.47 | 0.67–2.36 |
| Calculated corrosion margin (mm) with reference to pressure requirements | / | 0.93 | 0.93 | 0.93 |
| Service life (a) | 23.8 | 1.29–0.58 | 1.45–0.38 | 1.38–0.39 |

According to the corrosion rate in Figure 5 and the analysis in Table 6, it can be seen that when the average corrosion rates of 5Cr and 3Cr tubing materials reach the serious level under different CO₂ partial pressures in a simulation test environment within the range of $P_{CO2} \leq 1$ MPa, their service residual life cannot meet the design service life of the tubing string. In particular, when $P_{CO2} = 1$ MPa, the service life of two kinds of Cr-containing tubing materials is less than 1 year. However, the 9Cr material meets the design life requirements according to its corrosion rate calculation. So, the 9Cr material has high adaptability under this test environment while 5Cr and 3Cr have relatively poor adaptability.

## 4. Conclusions

1.  Under the simulated XX Oilfield formation water environment, the open circuit potentials of the three Cr-containing materials were 9Cr > 5Cr = 3Cr; the anodic polarization of the 9Cr material showed an obvious passivation zone, but the passivation state was unstable. Once the pitting corrosion was formed, it expanded continuously, and the pitting corrosion sensitivity was relatively high. The anodic polarization curves of 5Cr and 3Cr materials have no obvious passivation phenomenon.

2.  Under the condition of simulating the high temperature and high pressure formation water environment and $P_{CO_2}$ changing in the range of 0.2–1 MPa, the 9Cr material has a moderate or lower corrosion degree and relatively good corrosion resistance. The corrosion degrees of 5Cr and 3Cr materials were similar, and both of the two materials have extremely serious corrosion, without an obvious corrosion resistance advantage when compared with carbon steel N80.

3.  We found that the 9Cr material has good adaptability and meets the design service life of 10 years under the simulation test environment while the 5Cr and 3Cr tubing materials have poor adaptability. Therefore, it is not recommended to use 5Cr and 3Cr tubing materials without anticorrosion measures. We predict that 3Cr, 5Cr, and N80 materials will be used at lower partial pressures of $CO_2$ (<0.2 MPa).

**Author Contributions:** Methodology, G.L., J.L., Q.D. and Y.H.; Investigation, X.Z., J.L., M.L. and Q.D.; Resources, M.L. and Y.H.; Data curation, G.L.; Writing—original draft, X.Z.; Writing—review & editing, X.Z. All authors have read and agreed to the published version of the manuscript.

**Funding:** This work was supported by the National Science Foundation for Young Scientists of China under Grant (No. 51904331).

**Institutional Review Board Statement:** Not applicable.

**Informed Consent Statement:** Not applicable.

**Data Availability Statement:** Not applicable.

**Conflicts of Interest:** The authors declare no conflict of interest.

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
