# Peer review of "Corrosion Performance Analysis of Tubing Materials with Different Cr Contents in the CO2 Flooding Injection–Production Environment"

_coatings, doi:10.3390/coatings13101812_

Round 1
Reviewer 1 Report
The article "Corrosion Performance Analysis of Tubing Materials with Different Cr Contents in the CO2 Flooding Injection-Production Environment" appears to be well-structured and informative. However, I have some minor revision comments to improve the clarity and readability of the article:
1. The abstract is concise, but it could benefit from a clearer summary of the key findings. Additionally, it should mention the importance or implications of the research.
2. In the introduction, it's important to clearly state the research objectives or hypotheses. What specific questions is this study aiming to answer?
3. Consider providing a brief overview of the methodology used in the study.
4. Subdivide this section into subsections for better organization, such as "2.1 Test Material" and "2.2 Experimental Methods."
5. The table 1 is informative, but consider adding units (e.g., wt%) in the table header for clarity.
6. Include a brief caption that describes what the figure 1 is illustrating. What do the different colors or features represent?
7. Add a brief explanation of why the specific test conditions (temperature, scanning rate, etc.) were chosen. How do these conditions relate to real-world scenarios?
8. Consider providing a brief explanation of why these specific ions and concentrations were chosen for the test medium.
9. Provide a brief rationale for conducting the high-temperature and high-pressure immersion test. What real-world conditions does it simulate?
10. Check for consistent use of verb tenses throughout the article. For instance, ensure that past tense is used consistently when describing the methods and results.
11. In the conclusion section, summarize the key findings and their implications for practical applications more explicitly.
12. Consider revising the overall organization of the paper to ensure a logical flow of information from introduction to conclusion.
13. Consider adding a few more specific keywords that directly relate to the research topic.
14. There is lack of literature thus author is suggested to add some relevant literature such as: https://doi.org/10.3390/coatings6010012; 10.1088/1742-6596/2267/1/012079; https://doi.org/10.3390/coatings7120217; https://doi.org/10.3390/app13020730; https://doi.org/10.3390/polym12030689
Minor revision needed
Author Response
C
Coatings- ID 2627519
Dear reviewer :
Thank you for the comments on the manuscript, We have revised the manuscript according to your comments,and the detail responses to the comments of the editor are attached.
- The abstract is concise, but it could benefit from a clearer summary of the key findings. Additionally, it should mention the importance or implications of the research.
Answer: Thanks to the reviewer's suggestion. The author has improved the content of the abstract and supplemented the research purpose of this paper.
- In the introduction, it's important to clearly state the research objectives or hypotheses. What specific questions is this study aiming to answer?
Answer: Thanks to the reviewer's suggestion. The purpose of this study is mainly to understand the difference of corrosion properties of three kinds of low-Cr tubing materials and carbon steel in CO2 flooding and production environment of XX oilfield, to clarify the adaptation range of different materials, and to provide a strong theoretical basis for oil field material selection. The author adds the purpose and significance of this study in the preface.
- Consider providing a brief overview of the methodology used in the study.
Answer: Thanks to the reviewer's suggestion. Sections 2.2 and 2.3 of this paper describe the two simulation test methods used in the study, respectively. At the same time, the author improved the test method.
- Subdivide this section into subsections for better organization, such as "2.1 Test Material" and "2.2 Experimental Methods."
Answer: Thanks to the reviewer's suggestion. The author modified 2.1 and 2.2 according to the suggestions of the reviewer, and marked them in the paper.
- The table 1 is informative, but consider adding units (e.g., wt%) in the table header for clarity.
Answer: Thanks to the reviewer's suggestion. According to the suggestion of the reviewer, the author added the unit at the head of the table and marked it in the text.
- Include a brief caption that describes what the figure 1 is illustrating. What do the different colors or features represent?
Answer: Thanks to the reviewer's suggestion. Figure 1 shows the open circuit potential curves of four materials in a simulated environment. Different colors represent different materials. In the figure, different color lines represent the identification of which material, among which black is 9Cr, red is 3Cr, green is 5Cr, and blue is N80 material.
- Add a brief explanation of why the specific test conditions (temperature, scanning rate, etc.) were chosen. How do these conditions relate to real-world scenarios?
Answer: Thanks to the reviewer's suggestion. In this paper, the simulation test parameters are carried out according to the most harsh environment of the oilfield, the solution medium of the simulation test is prepared according to the ion concentration of the oilfield produced water environment, and the test temperature is the highest temperature of the oilfield working environment. The author has made improvements and modifications in the article, and made marks.
- Consider providing a brief explanation of why these specific ions and concentrations were chosen for the test medium.
Answer: Thanks to the reviewer's suggestion. The author has already answered the choice of medium in question 7. The solution medium of the simulation test is calculated according to the ion concentration of the produced water in the field, and the corresponding chemical reagent is prepared in the room with deionized water.
- Provide a brief rationale for conducting the high-temperature and high-pressure immersion test. What real-world conditions does it simulate?
Answer: Thanks to the reviewer's suggestion. High temperature and high pressure immersion test is to soak the sample in the simulated solution in the autoclave, then rise to the test temperature and fill CO2 to the required test pressure to realize the corrosion performance evaluation of the pipe in contact with the oil field produced water environment. The test simulates the temperature, pressure and solution medium of the pipe under the actual service condition, and the corrosion performance of the material under the synergistic action of multiple factors can be obtained.
- Check for consistent use of verb tenses throughout the article. For instance, ensure that past tense is used consistently when describing the methods and results.
Answer: Thanks to the reviewer's suggestion. The author checks the verb tenses in the article and makes some improvements and modifications.
- In the conclusion section, summarize the key findings and their implications for practical applications more explicitly.
Answer: Thanks to the reviewer's suggestion. In the conclusion, the author summarized the research results obtained from the experiment and clarified the using scope of the material.
- Consider revising the overall organization of the paper to ensure a logical flow of information from introduction to conclusion.
Answer: Thanks to the reviewer's suggestion. According to the suggestions of the reviewer, the author sorted out the chapter Settings of the article, adjusted the structure of the experimental part, and made marks in the article.
- Consider adding a few more specific keywords that directly relate to the research topic.
Answer: Thanks to the reviewer's suggestion. The author refined the abstract and added keywords related to the topic.
- There is lack of literature thus author is suggested to add some relevant literature such as: https://doi.org/10.3390/coatings6010012; https://doi.org/10.3390/coatings6010012; 10.1088/1742-6596/2267/1/012079; https://doi.org/10.3390/coatings7120217;
https://doi.org/10.3390/app13020730;
https://doi.org/10.3390/polym12030689
Answer: Thanks to the reviewer's suggestion. The author consulted the relevant articles suggested by the reviewer and supplemented them in the references.
Reviewer 2 Report
The article is devoted to a current topic, namely the study of the corrosion resistance of steel used in oil production and refining.
The article is written clearly and is easy to read.
I have several recommendations for paper authors:
1. To make it clearer for the reader how the High temperature and High pressure immersion test were carried out, please provide a diagram of the laboratory setup.
2. In Figures 2, 3, 4, 5, I recommend using the same colors of markers for 3Cr, 5Cr, 9Cr steel. Now the reader needs to be careful because the black marker indicates 3Cr in Figures 3 and 4, but 9Cr in Figures 1 and 5.
3. Photos of microstructures and oxides on the surface of the samples are of very good quality.
4. The conclusions from the study can be useful not only to other researchers, but also to industrial enterprises and oil producing companies.
5. The reference contains a lot of modern, relevant research.
Author Response
Coatings- ID 2627519
Dear reviewer:
Thank you for the comments on the manuscript, We have revised the manuscript according to your comments,and the detail responses to the comments of the editor are attached.
- To make it clearer for the reader how the High temperature and High pressure immersion test were carried out, please provide a diagram of the laboratory setup.
Answer: Thanks to the reviewer's suggestion. The author added a schematic diagram of the high temperature and high pressure immersion test device in the paper.
- In Figures 2, 3, 4, 5, I recommend using the same colors of markers for 3Cr, 5Cr, 9Cr steel. Now the reader needs to be careful because the black marker indicates 3Cr in Figures 3 and 4, but 9Cr in Figures 1 and 5.
Answer: Thanks to the reviewer's suggestion. In each diagram, the author lists the identification of the material represented by the color (such as dotted lines), and the author feels that it is not necessary for all diagrams to use the same color to represent the same material. For example, in Figure 4, there are two curves of the same material, if the same color is used, the reader will be confused. Therefore, the color identification in each diagram can guide the reader to distinguish the material.
- Photos of microstructures and oxides on the surface of the samples are of very good quality.
Answer: Thanks very much for the reviewer's encouragement and recognition, the author will continue to do better in the future.
- The conclusions from the study can be useful not only to other researchers, but also to industrial enterprises and oil producing companies.
Answer: Thank you very much for the reviewer's recognition to the author's work.
- The reference contains a lot of modern, relevant research.
Answer: Thank you very much for the reviewer's recognition to the author's work.
Reviewer 3 Report
Comments for the MINOR revision:
1. In abstract: mention the tubing material type and Cr content values.
2. In abstract: Expand “EDS”?
3. Include the future scope in the abstract or in the conclusion.
4. The limitation of the currently considered tubing materials with Cr content should be included.
5. Avoid “abbreviated terms in the Keywords. Ex. CO2
6. Novelty details were poorly formulated and the back ground information need to be enhanced more with recently published articles (in 2023)
7. Tables 1 and 2: references were missing.
8. Justification on the selection of based material (N80) and Cr content (up to 9Cr) should be discussed.
9. Results and Discussion: need a proper results backup for the observed results. Make a comparison table with similar materials and other additives for the corrosion characteristics and show much the present material was best over other methods.
10. Results and Discussion: physical phenomena discussions behind the attained results should be enhanced.
Minor editing of English language required
Author Response
Coatings- ID 2627519
Dear reviewer :
Thank you for the comments on the manuscript, We have revised the manuscript according to your comments,and the detail responses to the comments of the editor are attached.
- In abstract: mention the tubing material type and Cr content values.
Answer: Thanks to the reviewer's suggestion. The author added the specific types of pipes in the abstract. The chemical composition of the test materials is shown in Table 1, and the Cr content of each material can be seen.
- In abstract: Expand “EDS”?
Answer: Thanks to the reviewer's suggestion. The author has perfected the full name of EDS.
- Include the future scope in the abstract or in the conclusion.
Answer: Thanks to the reviewer's suggestion. In the abstract and conclusion, the authors added the application range of 3Cr,5Cr and carbon steel N80.
- The limitation of the currently considered tubing materials with Cr content should be included.
Answer: Thanks to the reviewer's suggestion. Different Cr-containing materials have suitable application environment, and the suitable application range can be found through adaptability evaluation.
- Avoid “abbreviated terms in the Keywords. Ex.
Answer: Thanks to the reviewer's suggestion. The author modified the abbreviations in the keywords.
- Novelty details were poorly formulated and the back ground information need to be enhanced more with recently published articles (in 2023)
Answer: Thanks to the reviewer's suggestion. The author modified some structural designs and added some recently references.
- Tables 1 and 2: references were missing.
Answer: Thanks to the reviewer's suggestion. Table 1 is the chemical composition analysis of the test materials, and Table 2 is the ion composition of the formation water provided by the oilfield during the simulation test. The solution medium for the simulation test is prepared by using this ion group without reference.
- Justification on the selection of based material (N80) and Cr content (up to 9Cr) should be discussed.
Answer: Thanks to the reviewer's suggestion. In this paper, three Cr-containing materials were selected for comparison with N80, in order to compare the differences in the corrosion resistance of 3Cr,5Cr and N80, and also compared the differences of corrosion resistance of 3Cr,5Cr and 9Cr materials, rather than directly comparing the corrosion resistance of N80 and 9Cr.
- Results and Discussion: need a proper results backup for the observed results. Make a comparison table with similar materials and other additives for the corrosion characteristics and show much the present material was best over other methods.
Answer: Thanks to the reviewer's suggestion. In this paper, the corrosion properties of three Cr-containing materials and carbon steel N80 were compared under different test conditions, and it was cleared that the corrosion resistance of 3Cr and 5Cr was not better than N80 in a certain concentration range of CO2 environment, aiming to solve the material selection concerns of XX oilfield.
- Results and Discussion: physical phenomena discussions behind the attained results should be enhanced.
Answer: Thanks to the reviewer's suggestion. The author reviews the results and discussion of the article, and improves the analysis of the results.